# Glial and neuronal Semaphorin signaling instruct the development of a functional myotopic map for *Drosophila* walking

**Durafshan Sakeena Syed[1], Swetha B.M. Gowda[1,2], O Venkateswara Reddy[1], Heinrich Reichert[3], K VijayRaghavan[1]\***

[1]National Centre for Biological Sciences, Tata Institute of Fundamental Research, Bangalore, India; [2]Manipal University, Manipal, India; [3]Biozentrum, University of Basel, Basel, Switzerland

**Abstract** Motoneurons developmentally acquire appropriate cellular architectures that ensure connections with postsynaptic muscles and presynaptic neurons. In *Drosophila*, leg motoneurons are organized as a myotopic map, where their dendritic domains represent the muscle field. Here, we investigate mechanisms underlying development of aspects of this myotopic map, required for walking. A behavioral screen identified roles for Semaphorins (Sema) and Plexins (Plex) in walking behavior. Deciphering this phenotype, we show that PlexA/Sema1a mediates motoneuron axon branching in ways that differ in the proximal femur and distal tibia, based on motoneuronal birth order. Importantly, we show a novel role for glia in positioning dendrites of specific motoneurons; PlexB/Sema2a is required for dendritic positioning of late-born motoneurons but not early-born motoneurons. These findings indicate that communication within motoneurons and between glia and motoneurons, mediated by the combined action of different Plexin/Semaphorin signaling systems, are required for the formation of a functional myotopic map.

**\*For correspondence:** vijay@ncbs.res.in

## Introduction

Motoneurons are key elements in the neural networks that generate behavior in all nervous systems. They represent the final common path for convergence of processed information from motor control circuitry in the central nervous system and from sensory feedback circuitry in the periphery (*Sherrington, 1906*; *Pearson, 1993*). Moreover, as final common output channels, they represent exclusive control elements of muscle effectors that mediate behavioral action. For their correct function, motoneurons must acquire specific cellular architectures during development such that their input domains, usually manifest as dendrites, receive connections from appropriate pre-motor neurons, and their axonal output domains make connections to correct target muscle cells (*Jessell et al., 2011*; *Harris et al., 2015*; *Arber, 2012*). How motoneuron development is orchestrated to establish the appropriate and specific dendritic input and axonal output connectivity needed for behavioral action is a central and still important question in neural development (*Tessier-Lavigne and Goodman, 1996*; *Jefferis, 2006*; *Dasen and Jessell, 2009*).

Considerable insight into the mechanisms that control motoneuron structure has been obtained in the neurogenetic model system of *Drosophila*. In this system, the developmental processes involved in generating appropriate dendritic and axonal morphology of different types of larval (*Landgraf and Thor, 2006*; *Mauss et al., 2009*) and adult motoneurons have been studied extensively. For example, the specific architecture of the motoneurons that innervate the adult leg has been shown to depend on their lineage and birth order, in that the majority of the leg motoneurons are postembryonic lineal descendants of neuroblast 15, and these motoneurons manifest a birth

**eLife digest** Nerve cells enable us to both sense the world around us and to move about it. The nerves responsible for movement are called motor neurons. While one end of a motor neuron stimulates the muscle it is connected to, the other end receives signals from nerves in the spinal cord that relay messages about movement from the brain. Motor neuron connections in the spinal cord, or its equivalent in insects, the ventral nerve cord, are organized into an arrangement known as a myotopic map, which reflects the anatomical arrangement of the muscles in the body. Much remains to be learnt about how these maps form.

Syed et al. have investigated how the myotopic map develops for motor neurons in the legs of fruit flies by reducing the function of chosen genes in the ventral nerve cord and asking how this affects the myotopic map. The experiments disrupted a signaling system called the Semaphorin signaling pathway that guides motor neurons to the right target muscle and consists of different receptor-signaling molecule pairs.

By looking for flies with an abnormal walk and with disrupted motor neuron organization, Syed et al. identified receptor-signal pairs that guide motor neurons to different leg muscles. Specific receptor-signal pairs also guide the organisation of motor neurons in the ventral nerve cord. This guidance depends on when neurons are 'born'. While a receptor-signal pair targets early born neurons to one leg muscle, the same receptor-signal pair regulates a different aspect of guidance in late-born neurons. Cells called glia, which are related to neurons, also help to position the connections of late-born motor neurons in the ventral nerve cord.

Overall, the Semaphorin signaling system assists communication both within motor neurons and between glia cells and motor neurons during the formation of the myotopic map for leg motor neurons. These discoveries open new avenues of investigation into how else these cells communicate with each other to aid the development and organization of motor neurons.

order-specific neuroanatomical organization (*Truman et al., 2004*; *Brierley et al., 2009*; *Brierley et al., 2012*; *Baek and Mann, 2009*). In this lineage, early born motoneurons project their axons to proximal muscles in the leg segments and have dendritic arborization that extends toward the thoracic ganglion neuropile midline, whereas late born motoneurons innervate distal muscles in the leg segments and have dendritic arbors that are restricted to the lateral regions of the ganglionic neuropile. Thus, in terms of their central and peripheral nervous architecture, these leg motoneurons form a myotopic map (*Brierley et al., 2009*). Although this myotopic map-specific targeting of leg motoneuron dendrites has been shown to require midline signaling through the Slit/Roundabout and Netrin/Frazzled signaling systems (*Brierley et al., 2009*), the mechanism underlying compartmentalization in terms of birth order and distinct axonal and dendritic targeting is poorly understood. In order to understand the guidance cues responsible for leg motoneuron development, we performed a behavioral screen for locomotor defects caused by targeting RNAi in motoneurons and identified Plexin and Semaphorin as candidates. In this study, we have investigated the development of specific axonal projections and dendritic arbors in leg motoneurons required for walking in adult *Drosophila* mediated by Plexin/Semaphorin signaling system and, in doing so, discover novel cellular and molecular roles.

The Semaphorins (Sema) are a large family of transmembrane and secreted glycoprotein ligands that together with their Plexin (Plex) receptors are known to be involved in control of cell migration, dendritic topography and axon guidance in vertebrates and invertebrates (*Kolodkin et al., 1993*; *Yazdani and Terman, 2006*; *Pasterkamp, 2012*). Semaphorins and Plexins have evolutionarily conserved guidance function during nervous system development, and both transmembrane and secreted Semaphorin ligands can mediate a diverse set of repulsive and attractive guidance functions. In *Drosophila*, there are two Plexin receptors and five Semaphorin ligands. PlexinA (PlexA) strongly binds the Sema1a and Sema1b ligands while as Plexin B (PlexB) binds to Sema2a and Sema2b (*Bates and Whitington, 2007*; *Ayoob et al., 2006*; *Lattemann et al., 2007*; *Sweeney et al., 2011*) (Sema5c is not expressed in the CNS). Both PlexA and PlexB have been shown to mediate a Sema-dependent repulsion of motor axons during embryonic development of

body wall innervation (*Winberg et al., 1998*; *Yu et al., 1998*; *Jeong et al., 2012*). Whether or not Plexin/Semaphorin signaling is also involved in the development of the post-embryonically generated motoneurons that innervate the leg musculature is currently unknown.

We first carry out a behavioral screen for locomotor defects caused by targeted RNAi knockdown of Semaphorins and Plexins in leg motoneurons, which shows that knockdown of membrane-bound Plexins and Semaphorins results in abnormal walking gait. Based on this, we screen for corresponding neuroanatomical defects in the affected leg motoneurons. We find that motoneuron-specific knockdown of Sema1a or PlexA causes defective axonal defasciculation and targeting in motoneurons that innervate leg muscles and these phenotypes differ in proximal femur and distal tibia. In the femur, there occurs a reduction in axon branching and defasciculation of motoneurons, whereas an increase in axon branching of motoneurons occurs in the tibia. This suggests a compartment-specific activity of Plexin/ Semaphorin signaling system.

Larval and adult motoneuron dendrites in *Drosophila*, organized as a myotopic map, have been shown to utilize Robo-Slit and Netrin-Frazzled signaling systems (*Brierley et al., 2009*; *Mauss et al., 2009*). The dendritic targeting of larval motoneurons is an active process that is independent of glial differentiation or target muscle formation (*Landgraf et al., 2003*; *Landgraf and Thor, 2006*). In contrast, we find the role of glia in positioning dendrites of late born motoneurons that innervate distal muscles of tibia and occupy lateral regions of the thoracic neuropil in adult *Drosophila*. We demonstrate that motoneuron-specific knockdown of PlexB or knockdown of Sema2a secreted from glial cells results in a shift of dendritic arborization toward the ganglionic midline in late born leg motoneurons. This is an important and novel difference, which may have general implications for dendritic patterning mechanisms.

These findings indicate that the integrative action of multiple Plexin/Semaphorin signaling systems mediate communication between glia and motoneurons and within motoneurons for the correct formation of axonal projections and dendritic arborization in leg motoneuron development.

## Results

### Motoneuron-specific knockdown of Plexin/Semaphorin signaling results in locomotor defects

To determine which elements of the Plexin/Semaphorin signaling system might be required for correct leg locomotor activity, we first carried out a behavioral screen for walking defects caused by targeted RNAi knockdown of Semaphorins and Plexins in leg motoneurons. In these experiments, the OK371-Gal4 driver was used to target UAS-RNAi knockdown to leg motoneurons; this Gal4 driver targets reporter gene expression to all motoneurons during embryonic and postembryonic development and in the adult (*Mahr and Aberle, 2006*; *Brierley et al., 2009*; *Brierley et al., 2012*). In these experiments, UAS-RNAi knockdown constructs for Sema1a and its receptor PlexA as well as for Sema2a and its receptor PlexB were used. To monitor their locomotor activity, flies were allowed to walk freely over a soot plate (*Maqbool et al., 2006*) and their resulting footprint patterns were documented and analyzed in knockdown versus control flies.

Locomotor defects were observed in RNAi-mediated knockdown of Sema1a, PlexA, and PlexB in leg motoneurons (*Figure 1*). In Sema1a knockdown experiments, flies showed uncoordinated walking activity with leg dragging. In PlexA knockdown experiments, flies showed uncoordinated walking, short steps or short steps with leg dragging. In PlexB knockdown experiments, flies walked with short steps and leg dragging. In Sema2a knockdown experiments, flies showed normal walking behavior. The frequency of occurrence of phenotypes is shown in *Figure 1—source data 1*. Taken together, these results suggest that PlexA, PlexB, and Sema1a, but not Sema2a, are required in leg motoneurons during development and/or in the adult for the manifestation of correct locomotor activity.

Given the previously demonstrated roles of Plexin/Semaphorin signaling in the development of appropriate dendritic and axonal morphology in other neuronal systems (and during embryogenesis), we next investigated if these defects in locomotor activity might be predictive of knockdown-induced axonal and/or dendritic projection defects in the targeted leg motoneurons. We first focused on PlexA and its transmembrane ligand Sema1a.

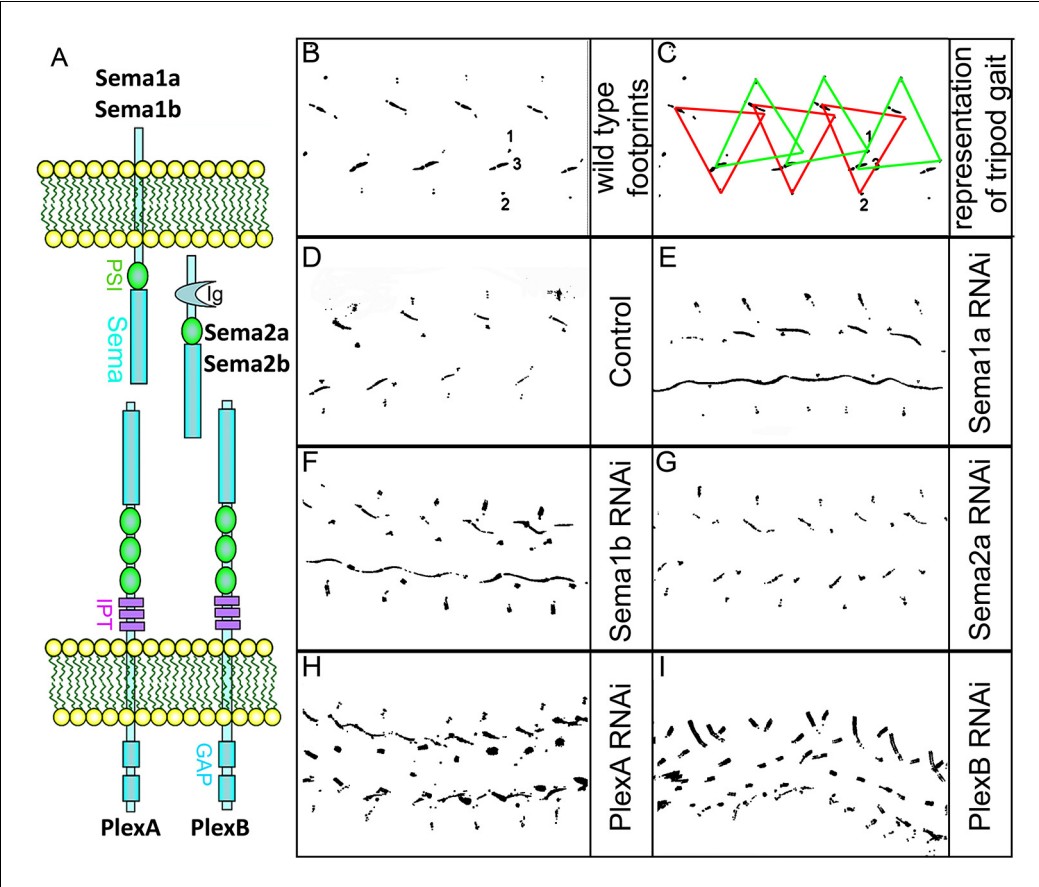

**Figure 1.** Targeted knockdown of Plexin/Semaphorin signaling in motoneurons results in walking defects. (**A**) Schematic of interactions between Sema1a, Sema1b, and PlexA as well as between Sema2a, Sema2b, and PlexB. (**B–I**) Walking patterns monitored by footprint tracking in control and Plexin/Semaphorin RNAi knockdowns targeted to motoneurons. (**B**, **C**) Wild-type walking pattern. 1, 2 and 3 denote footprints of first, second, and third leg, respectively. Colored triangles denote the footprints of legs in the stance phase, where three legs are on the ground at any given time, representing tripod gait. The front leg and hind leg from one side move together with the middle leg on the opposite side. This alternates when the fly takes a step forward, represented by red and green triangles. Defective walking patterns are observed in (**E**) Sema1a, (**F**) Sema1b, (**H**) PlexA, (**I**) PlexB, but not in (**G**) Sema2a knockdowns.

The following source data is available for figure 1:

**Source data 1.** Summary of walking behavior upon Plexin/Semaphorin knockdown in motoneurons.

## PlexA and Sema1a are required for correct axonal projections of leg motoneurons

To determine if PlexA and Sema1a are required for correct targeting of motoneuron axons to specific muscles in the leg, we knocked down these signaling molecules in all the leg motoneurons using the OK371-Gal4 driver and examined fasciculation and targeting of their axons in the adult leg periphery. Effects on the motoneuronal innervation onto the proximal muscles of the femur and the distal muscles of the tibia were characterized separately (*Figure 2—figure supplement 1*).

In the wild-type femur, the proximal axon branch defasciculates from the main motor nerve and innervates the ltm2 (long tendon muscle 2), while as another axon branch defasciculates from the main motor nerve more distally, projects further distally and innervates the tidm (tibia depressor muscle) (*Figure 2A–D*). Following targeted knockdown of either PlexA or Sema1a, axon projection defects were seen in the proximal femur; for both PlexA and Sema1a knockdowns two similar kinds of projection phenotypes were observed either separately or in combination (*Figure 3—figure*

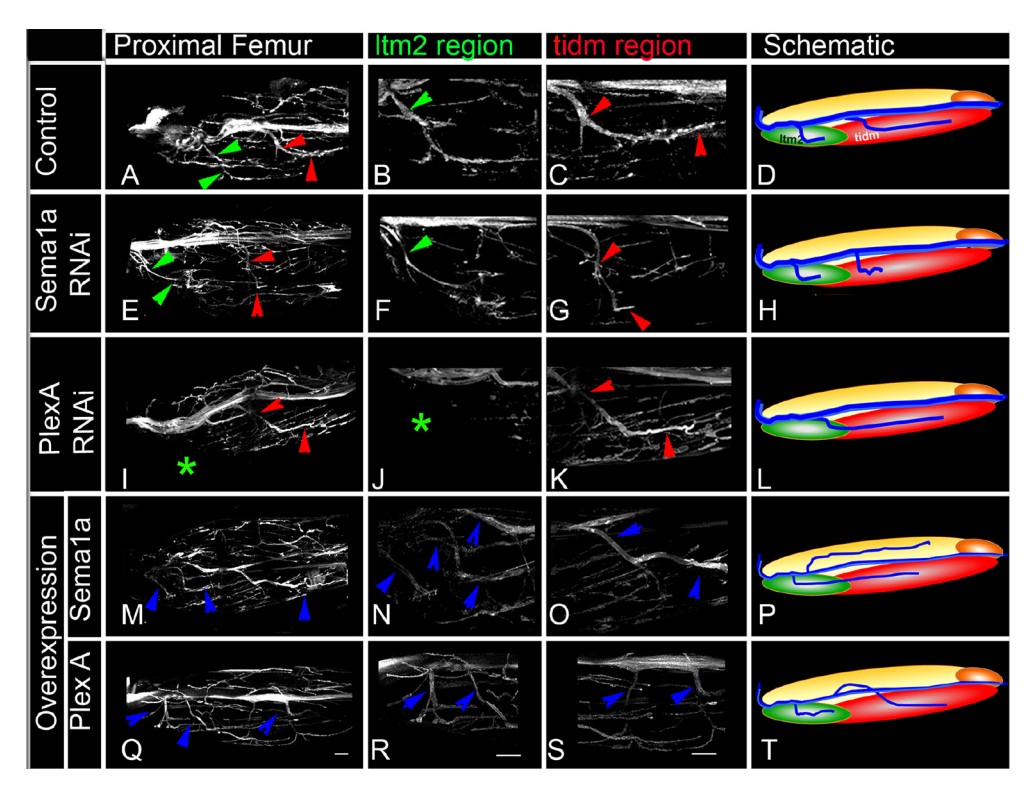

**Figure 2.** PlexA and Sema1a are required for correct axonal projections of leg motoneurons in the proximal femur. Targeted knockdown, overexpression, and labeling mediated by motoneuron-specific OK371-Gal4 driver. (**A–D**) Control innervation of femur. Axon projection defects characterized by decreased innervation are observed in (**E–H**) Sema1a knockdown and (**I–L**) PlexA knockdown. Extensive defasciculation and ectopic branches exiting the main motor nerve are observed in (**M–P**) Sema1a overexpression and (**Q–T**) PlexA overexpression. (**A, E, I, M, Q**) show overview of proximal femur innervation. (**B, F, J, N, R**) show magnified views of innervation in ltm2 (long tendon muscle 2) region. (**C, G, K, O, S**) show magnified views of innervation of proximal tidm (tibia depressor muscle). (**D, H, L, P, T**) show schematic summaries of femur innervation of these two nerves. ltm2 muscle in schematic marked by green and tidm in red. The main nerve innervating ltm2 region is outlined by green arrowheads; main nerve innervating tidm outlined by red arrowheads. Green asterisk denotes absence of innervation. Multiple nerves innervating ltm2 and tidm in over-expression outlined by blue arrowheads. Scale bars = 20 microns.

The following figure supplements are available for figure 2:

**Figure supplement 1.** Axonal innervation of motoneurons in the leg.

**Figure supplement 2.** Plex A and Sema1a are required for correct axonal fasciculation of leg motoneurons in the proximal femur.

*supplement 1A*; *Figure 3—source data 1*). In the first phenotype, the axon branch that normally innervates the tidm defasciculates from the main motor nerve correctly, but then stalls and does not project distally along the femur to innervate the tidm correctly (*Figure 2E–H*). In the second phenotype, the axon branch that normally innervates ltm2 fails to defasciculate and exit from the main motor nerve and innervation of the ltm2 is lacking (*Figure 2I–L*).

In the distal wild-type tibia, several axon branches defasciculate from the main motor nerve to innervate the tadm (tarsus depressor muscle) and two terminal branches defasciculate from the main motor nerve to innervate the tarm (tarsus reductor muscle) (*Figure 3A–D*). After targeted knockdown of PlexA, axon projection defects characterized by a marked increase in the number of axon branches that exit the main nerve resulted (*Figure 3 E-H*). This increase was seen both in the axon

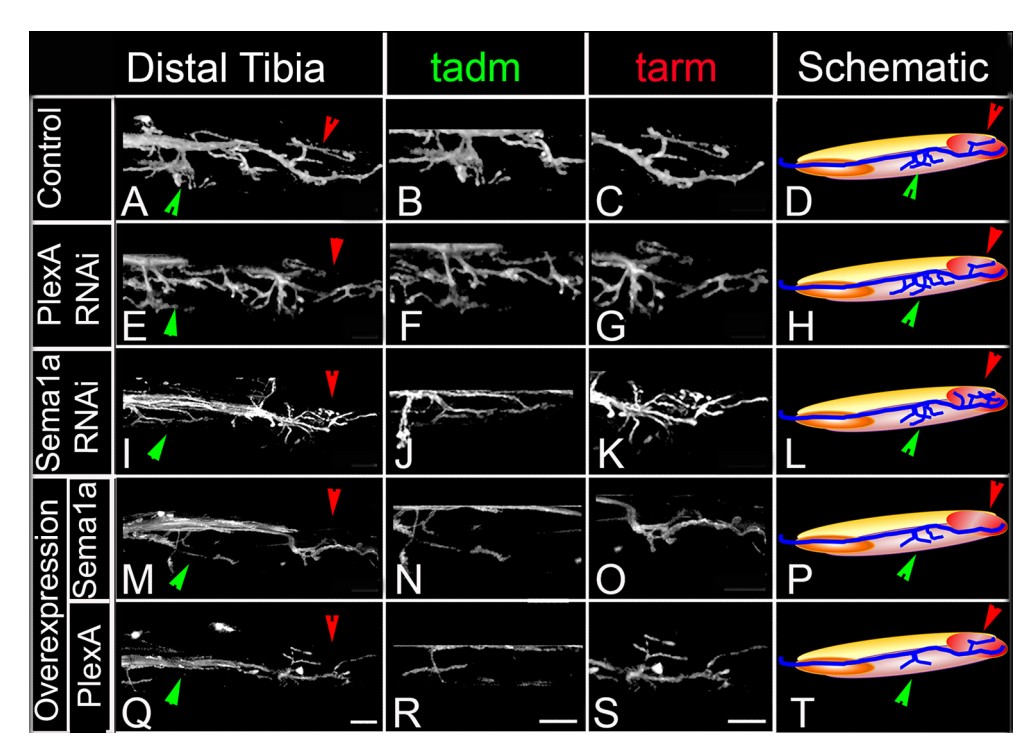

**Figure 3.** PlexA and Sema 1a are required for correct axonal projections of leg motoneurons in the distal tibia. Targeted knockdown, overexpression, and labeling mediated by motoneuron-specific OK371-Gal4 driver. (**A–D**) Control innervation of distal tibia. Axon projection defects characterized by increased innervation observed in (**E–H**) PlexA RNAi knockdown and (**I–L**) Sema1a RNAi knockdown. Decreased defasciculation of axonal branches exiting the main motor nerve are observed in (**M–P**) Sema1a overexpression and (**Q–T**) PlexA overexpression. (**A, E, I, M, Q**) show overviews of distal tibia innervation. (**B, F, J, N, R**) show magnified views of innervation of tadm (tarsus depressor muscle). (**C, G, K, O, S**) show magnified views of innervation of tarm (tarsus reductor muscle). (**D, H, L, P, T**) show schematic summaries of tibia innervation. Green arrowheads point toward innervations in tadm and red arrowheads point toward tarm. Scale bars = 20 microns.

The following source data and figure supplement are available for figure 3:

**Source data 1.** Summary of the axon defasciculation and targeting phenotypes in proximal femur and distal tibia. File contains underlying source data for *Figure 3—figure supplement 1*.

**Figure supplement 1.** Summary of the axon defasciculation and targeting phenotypes mediated by Sema1a and PlexA, in ways that differ in the proximal femur and distal tibia.

branches that innervate the tadm and in the axon branches that innervate the tarm. A comparable increase in the number of axon branches that exit the main nerve resulted after targeted knockdown of Sema1a; however, this was usually more pronounced in the axon branches that innervate tarm (*Figure 3I–L*; *Figure 3—figure supplement 1B*).

Taken together, these findings indicate that an abnormal reduction of PlexA/Sema1a in all leg motoneurons can lead to defective axonal branching phenotypes. The observed phenotypes differ in the proximal femur and the distal tibia in that a reduction in axon branching of motoneurons occurs in the femur and an increase in axon branching of motoneurons occurs in the tibia.

## Overexpression of PlexA and Sema1a result in aberrant axon projections of leg motoneurons

Given the axon branching phenotypes observed following an abnormal reduction of PlexA/Sema1a in leg motoneurons in the knockdown experiments, we next investigated if an abnormal increase of

PlexA/Sema1a in motoneurons might also result in aberrant axonal projection phenotypes. For this, we used the OK371-Gal4 driver to overexpress Sema1a and PlexA in all leg motoneurons.

In the proximal femur, targeted overexpression of Sema1a in leg motoneurons resulted in an increase in axon branching that was characterized by excessive defasciculation and ectopic branches exiting the main motor nerve in all preparations analyzed (*Figure 2M–P*). Moreover, this extensive defasciculation was accompanied by a correspondingly reduced thickness of the nerve's main axon bundle (*Figure 2—figure supplement 2*). Comparable defasciculation phenotypes were observed in targeted PlexA overexpression experiments (*Figure 2Q–T*; *Figure 3—figure supplement 1C*). However, in most cases, these defasciculation phenotypes were less extensive than in the Sema1a overexpression experiments, and a marked reduction of thickness in the main axon bundle of the nerve was usually not seen (*Figure 2—figure supplement 2*).

In the distal tibia, targeted overexpression of PlexA in leg motoneurons resulted in a decrease in axon branching that was characterized by a striking reduction in the number of axon branches that exit the main motor nerve and innervate the tadm and tarm (*Figure 3Q–T*). Targeted overexpression of Sema1a also resulted in reduced defasciculation and branching phenotypes in several cases; however, these were less extensive than in the PlexA overexpression experiments (*Figure 3M–P*; *Figure 3—figure supplement 1D*; *Figure 3—source data 1*).

Thus, both reduction and increase of PlexA/Sema1a expression in motoneurons can result in aberrant axonal branch formation in the proximal muscles of the femur and in the distal muscles of the tibia. Moreover, reduction and increase in PlexA/Sema1a expression result in opposing branching phenotypes. Taken together, these experiments indicate that a normal level of PlexA/Sema1a-mediated signaling is required in leg motoneurons for the formation of correct axonal projections and targeted muscle innervation in the leg.

## PlexA and Sema1a are required in femur motoneurons to maintain appropriate- and prevent inappropriate-innervation

The experiments reported above indicate that the targeted knockdown of PlexA/Sema1a in leg motoneurons result both in reduced axonal branching in the femur and increased axonal branching in the tibia. The coupled occurrence of these two phenotypes could arise, at least in part, if motoneurons that innervate the femur in the wild type mis-project into the tibia following PlexA/Sema1a knockdown. To investigate this possibility, we used the VGN6341-Gal4 driver which targets knockdowns to 3–4 motoneurons that innervate the femur as well as 1–2 motoneurons that innervate the tibia.

In the wild type, femur motoneurons labeled by VGN6341-Gal4 always innervated the tidm and ltm2 muscles of the proximal femur and, similarly, tibia motoneurons always innervated the tadm muscles of the distal tibia. In contrast, following targeted knockdown of PlexA and Sema1a, motoneurons that normally innervated the femur manifested a significant number of mis-projection phenotypes characterized by lack of innervation of the tidm and ltm2 muscles of the femur and ectopic innervation of the ltm1 and tidm muscles of the tibia. Comparable mis-projection phenotypes were seen in these femur motoneurons in a Sema1a homozygous null mutant, Sema1a hetreroallelic combination and in animals trans-heterozygous for mutants in both Sema1a and PlexA of the genotype Sema1a$^{P2}$/+;;PlexA$^{09}$/+. *Figure 4* and *Figure 4—figure supplement 1A*; *Figure 4—source data 1* show the range of phenotypes. We used another Gal4 driver, R6B011 which targets knockdowns to 3–4 motoneurons that innervate the femur and 2–3 motoneurons that innervate the tibia. In the wild type, femur motoneurons labeled by R6B011-Gal4 always innervated the proximal femur. In contrast, following targeted knockdown of PlexA, motoneurons that normally innervated the femur manifested a significant number of mis-projection phenotypes characterized by lack of innervation of the tidm and ltm2 muscles of the femur. Similar but weaker phenotype was observed following targeted knockdown of Sema 1a (*Figure 4—figure supplement 1B*, *Figure 4—figure supplement 2*).

To investigate if tibial motoneurons mis-project processes into the femur following PlexA/Sema1a knockdown, we used the VGN9281-Gal4 driver which targets knockdowns to a single motor neuron that innervates the tarsus reductor muscle (tarm2). Interestingly, while the targeted motorneuron did not misproject into femur, it showed abnormally increased branching in the distal tibia (*Figure 4—figure supplement 1C*, *Figure 4—figure supplement 3*). The ltm2 muscle in proximal femur is innervated by early born motoneurons and the tadm/ tarm muscles in the distal tibia are innervated by late born motoneurons (*Brierley et al., 2009*; *2012*).

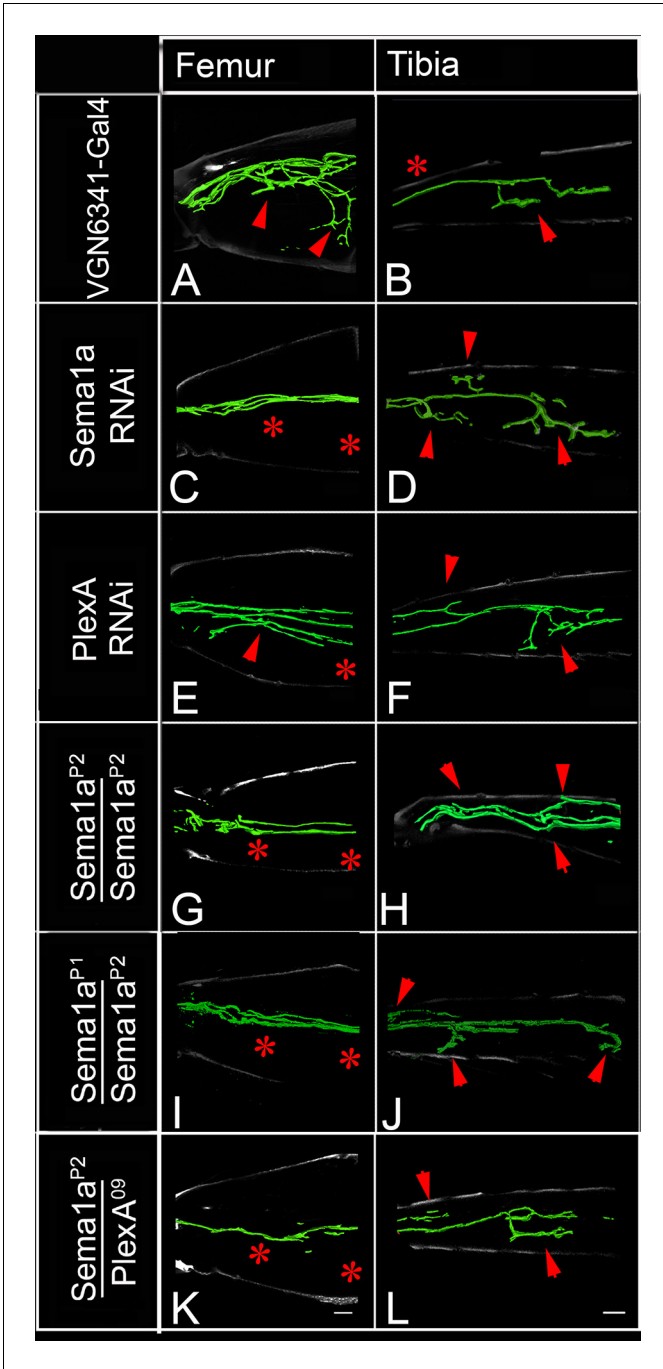

**Figure 4.** PlexA and Sema1a are required in femur motoneurons to maintain appropriate innervation of the femur and prevent inappropriate innervation of the tibia. VGN6341-Gal4 targeted knockdown of PlexA and Sema1a in a small subset of leg motoneurons. (**A, C, E, G, I, K**) innervation of proximal femur. (**B, D, F, H, J, L**) innervation of tibia. (**A, B**) Control innervation by targeted motoneurons. (**C, D**) Sema1a knockdown in targeted motoneurons. (**E, F**) PlexA knockdown in targeted motoneurons. (**G, H**) Sema1a$^{P2}$ homozygous mutant. (**I, J**) Sema1a$^{P1}$/Sema1a$^{P2}$ heteroallelic combination. (**K, L**) Sema1a$^{P2}$/+;;PlexA$^{09}$/+ trans-heterozygous mutant combination. Decreased innervation occurs in femur (**C, E, G, I, K**) and ectopic innervation occurs in tibia (**D, F, H, J, L**). Arrowheads point toward innervation and asterisk denote absence of normal innervation. Scale bars = 20 microns.

The following source data and figure supplements are available for figure 4:

**Source data 1.** Summary of the motoneuron axon defasciculation and targeting phenotypes in early and late-born motoneurons. File contains underlying source data for *Figure 4—figure supplement 1*.

*Figure 4 continued on next page*

*Figure 4 continued*

**Figure supplement 1.** Summary of the motoneuron axon defasciculation and targeting phenotypes mediated by Sema1a and PlexA in early and late-born motoneurons.
**Figure supplement 2.** PlexA and Sema1a are required in femur motoneurons for axonal targeting and defasciculation.
**Figure supplement 3.** Plex A and Sema1a are not required in tibia late-born motoneurons for axonal targeting.

Taken together, these findings imply that a reduction of PlexA/Sema1a in motoneurons that normally innervate the femur can result in a loss of appropriate femoral innervation and a gain of inappropriate tibial innervation. These specific mis-projection phenotypes of femur motoneurons are likely to contribute to the observed overall axonal projection phenotype in the leg, characterized by decreased axonal branching in the femur and increased axonal branching in the tibia.

## Sema2a but not Sema1a is highly expressed in the ganglionic midline

Previous work on the development of motoneurons during embryogenesis has shown that both the transmembrane Sema1a ligand acting through PlexA, and the secreted Sema2a ligand acting through PlexB are important for correct axonal outgrowth (*Matthes et al., 1995*; *Yu et al., 1998*; *Winberg et al., 1998*). The absence of locomotor defects in our motoneuron-targeted Sema2a knockdown experiments (see above) suggests that Sema2a might not act in an autonomous manner on leg motoneuron development. Indeed, motoneuron-specific knockdown of Sema2a using the OK371-Gal4 driver did not result in peripheral axon projection defects in leg motoneurons (data not shown). To investigate the possibility that Sema2a might act on leg motoneurons in a non-cell autonomous manner, we first compared the expression pattern of Sema1a and Sema2a in the thoracic ganglia during postembryonic development using immunolabeling.

Throughout postembryonic development, both Sema1a and Sema2a are widely expressed throughout most of the developing thoracic ganglia (*Figure 5A–F*). In this respect, their expression is comparable to that of their PlexA and PlexB receptors, which are also broadly expressed in the nervous system (*Winberg et al., 1998*). However, Sema1a and Sema2a expression levels are significantly different at the ganglionic midline. Thus, whereas Sema1a expression decreases toward the midline and has its lowest level of expression there, Sema2a expression is highest at/and near midline where a peak in its expression level is manifest (*Figure 5G–H*; *Figure 5—figure supplement 1*). This high level of expression in and around midline cells, many of which are glial in nature, suggests that Sema2a ligand secreted from these cells could affect leg motoneuron development within the thoracic ganglia and hence, influence the pattern of dendrite outgrowth in leg motoneurons in a non-cell autonomous manner.

## PlexB and Sema2a are required for correct dendritic targeting of leg motoneurons

To determine if Sema2a and PlexB are required for correct central dendritic arborization of leg motoneurons, we carried out RNAi knockdown experiments and examined the dendritic projection patterns of motoneurons in the thoracic ganglia. For this, we analyzed the dendrites of two sets of motoneurons labeled by dye backfilling techniques (see 'Materials and methods'). Dye backfilling of leg nerve from proximal muscles of femur labeled early-born motoneurons while backfilling the leg nerve from distal muscles of the tibia labeled late-born motoneurons (*Figure 6A–D*). In the wild type, early-born motoneurons have dendrites that arborize primarily in the ipsilateral neuropile, but also project one prominent branch to the ganglionic midline (*Figure 6C*). In contrast, wild-type late-born motoneurons have dendrites whose arborizations are restricted to lateral part of the ipsilateral neuropile; none of their dendrites project medially (*Figure 6D*).

RNAi knockdown of PlexB in motoneurons using the OK371-Gal4 driver resulted in a marked dendritic arborization phenotype in the late-born motoneurons but not in the early born motoneurons (*Figure 6E,F*). This phenotype was characterized by a shift of the dendritic arborizations toward the

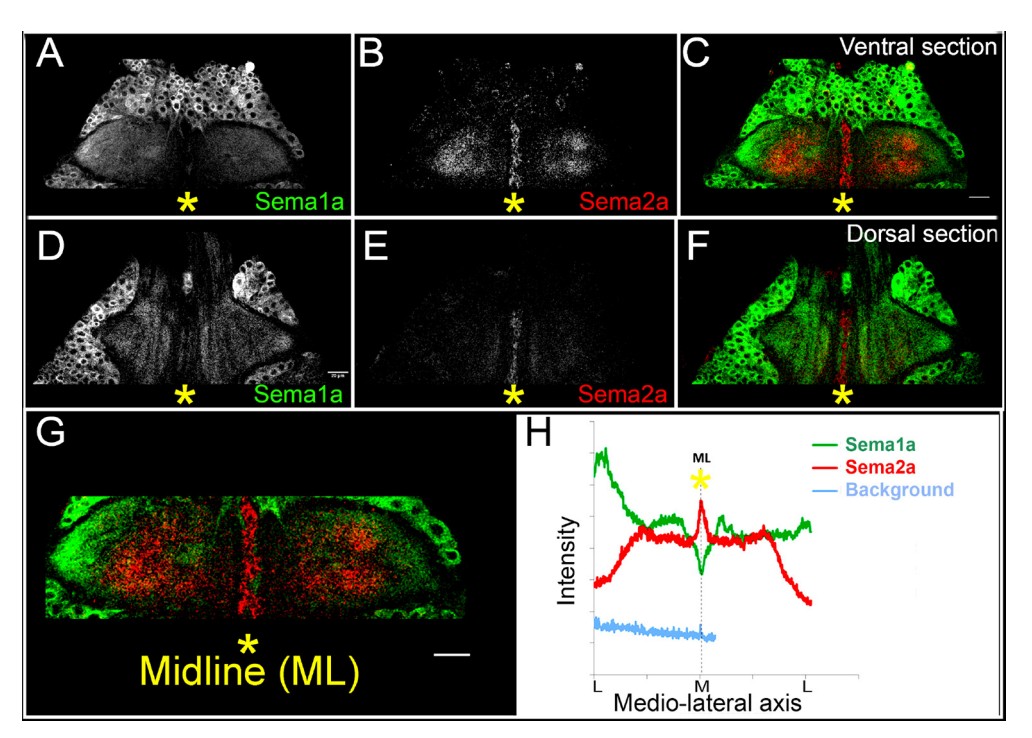

**Figure 5.** Sema2a is highly expressed in the ganglionic midline and intermediate region. Immunocytochemical analysis of expression of Sema1a and Sema2a in prothoracic ganglion at 25h APF (after puparium formation). (A–C) Ventral section. (D–F) Dorsal section. (A, D) Sema1a immunolabeling. (B, E) Sema2a immunolabeling. (C, F) Overlay of Sema1a and Sema2a immunolabeling. (G) Magnified view of neuropile. (H) Intensity profile of Sema1a and Sema2a immunolabeling taken along mediolateral axis of neuropile (in Z-stack; overlay of all optical sections). Yellow asterisk is placed at the ganglionic midline (ML). Sema2a level is highest at the midline and high in the intermediate regions in the neuropile and very low at the lateral edge. Scale bar = 20 microns.

The following figure supplement is available for figure 5:

**Figure supplement 1.** Dorso-ventral view of Sema2a expression in the thoracic neuropil.

---

midline as well as by the formation of a prominent ectopic dendritic branch, which projects to the ganglionic midline. In this respect, late-born neurons in the knockdown experiment resemble early-born neurons in the wild type. To confirm this dendritic arborization phenotype, we used the VGN9281-Gal4 driver line to target PlexB knockdown to specific late-born motoneurons (see *Brierley et al., 2009*). This resulted in the expected dendritic arborization phenotypes in targeted motoneurons, in that these late-born motoneurons, which normally restrict their dendritic arbors to lateral domains, mis-projected major dendritic branches toward the ganglionic midline (data not shown).

Targeted knockdown of Sema2a in leg motoneurons using the OK371-Gal4 driver had no effect on dendritic arbors of late-born or early-born motoneurons (data not shown). This corresponds to the lack of locomotor defects in behavioral experiments in which Sema2a RNAi was targeted to leg motoneurons (see *Figure 1*) and supports the notion that Sema2a might not act cell autonomously in motoneurons. Since immunolabeling studies showed that Sema2a expression in the neuropile is most prominent in/and near midline (glial) cells (see *Figure 5*), we carried out a targeted knockdown of Sema2a in glial/midline cells using either the glial-specific Repo-Gal4 driver or the midline-specific Slit-Gal4 driver.

In both cases, Sema2a knockdown resulted in clear dendritic arborization defects of leg motoneurons. Moreover, these defects were observed in late-born motoneurons but not in early-born motoneurons. Thus, following targeted knockdown of Sema2a in glial cells, late-born motoneurons,

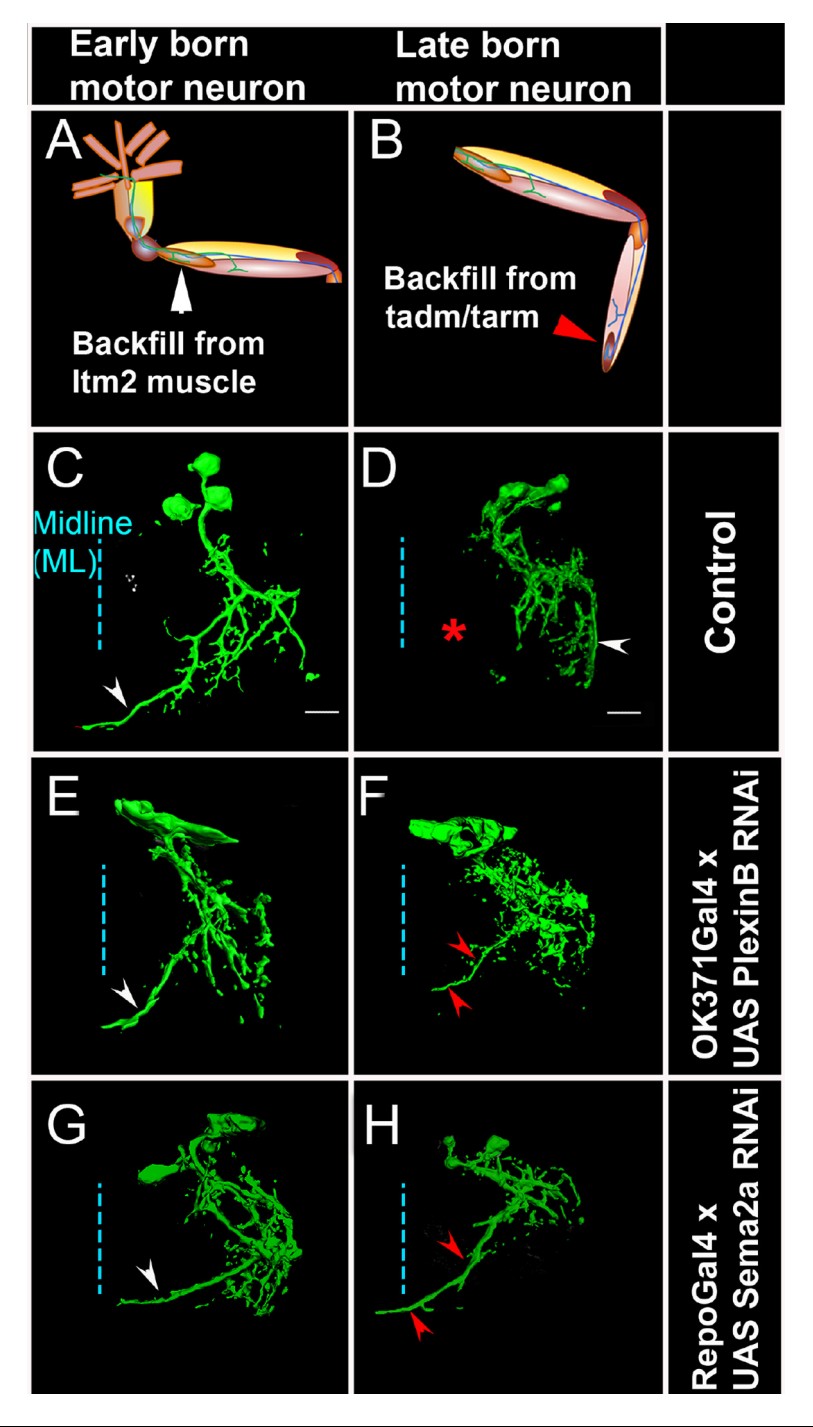

**Figure 6.** PlexB and Sema2a are required for correct dendritic targeting of leg motoneurons. Motoneuron dendrites in prothoracic hemiganglion labeled by dye backfilling from innervated muscles. (A, C, E, G) early born motoneurons that innervate ltm2. (B, D, F, H) late born motoneurons that innervate tadm. (A, B) Schematic representation of dye backfilled leg muscles. (C, D) Dendrites of motoneurons in wild-type control. (E, F) Dendrites of motoneurons following motoneuron-specific targeted knockdown of PlexB. (G, H) Dendrites of motoneurons following glial-cell-specific targeted knockdown of Sema2a. A dendritic mis-projection phenotype occurs in late-born motoneurons (F, H) but not in early-born motoneurons (E, G). White arrowhead indicates normal dendritic projection and red arrowhead indicates mis-projection. Red asterisk marks the region where projection is normally absent in late-born neurons. Scale bars =20 microns.

*Figure 6 continued on next page*

*Figure 6 continued*

The following figure supplements are available for figure 6:

**Figure supplement 1.** Sema2a secreted by midline cells is required for dendritic targeting of late-born motoneurons.

**Figure supplement 2.** Plex B/Sema2a is required for correct dendritic targeting of late-born motoneurons.

which normally restrict their dendritic arbors to lateral domains, shift their dendritic arbors medially and mis-project major dendritic branches toward the ganglionic midline (*Figure 6G,H*; *Figure 6—figure supplement 1*). Comparable mis-projection phenotypes were seen in dendrites of late born motoneurons in animals of the genotype Sema2a/+; PlexB/+ (*Figure 6—figure supplement 2*). Upon this dendritic mis-projection in late-born motoneurons, dendritic arborizations of these motoneurons acquired morphological characteristics of arborizations of normal early born neurons.

To investigate if overexpressing Sema1a (or Sema2a) in the midline repels dendritic arbors of early born motoneurons from the midline, with these motor neurons acquiring morphological characterists of late-born neurons, we carried out a targeted over-expression of Sema1a (or Sema2a) using the midline-specific Slit-Gal4 driver. In this case, the dendritic arborization of early born neurons appeared to be normal (data not shown).

Taken together these experiments indicate that PlexB/Sema2a signaling is required for the formation of correct dendritic projections of late-born leg motoneurons. Moreover, they imply that the requirement in these motoneurons is cell autonomous for PlexB and non-cell autonomous (i.e. glial/ midline cell-derived) for Sema2a.

## Discussion

In this report, we analyze the role of the Plexin/Semaphorin signaling system in the formation of appropriate axonal and dendritic structures in the adult-specific leg motoneurons of Drosophila. Our findings reveal multiple roles of different Semaphorins and Plexins in axon guidance in the periphery and in dendritic targeting in the central neuropile for these motoneurons. In the following, we discuss the major implications of these findings.

### Plexins and Semaphorins in motoneuron axon guidance

During development, motoneurons extend axons over relatively long distances from the thoracic ganglia to the numerous muscles located within their peripheral targets. During this process, multiple guidance cues act to ensure that the motoneurons establish their complex innervation patterns with high precision (*Tessier-Lavigne and Goodman, 1996*; *Raper and Mason, 2010*; *Kolodkin and Tessier-Lavigne, 2011*). As motoneurons navigate into the periphery, they often initially fasciculate with each other to form nerve branches and, once they reach their general target regions, defasciculate and exit their nerve branches to innervate specific muscles. Repulsive Semaphorin signaling through Plexin receptors has been implicated in this process in vertebrates and invertebrates (*Pasterkamp et al., 2006*; *2012*). In *Drosophila*, Plexin/Semaphorin signaling has been studied in the development of the embryonically generated motoneurons that innervate the larval body wall (*Winberg et al., 1998*; *Yu et al., 1998*; *Ayoob et al., 2006*). Sema1a acting through PlexA has been shown to be important for the regulation of motoneuron axon defasciculation through axon-axon repulsion (*Winberg et al., 1998*; *2001*; *Yu et al., 1998*; *2000*). Similarly, Sema2a acting through PlexB is required for correct motoneuron axon defasciculation, but also has additional functions in motoneuron axon navigation (*Matthes et al., 1995*; *Ayoob et al., 2006*) implying that PlexA/ Sema1a and PlexB/Sema2a have both shared and distinct function in motor axon guidance.

Our analysis of axonal targeting in leg motoneurons indicates that the PlexA/Sema1a signaling system is also required in postembryonic development of adult-specific motoneurons. Targeted knockdown of PlexA or Sema1a in leg motoneurons results in prominent defasciculation and targeting defects of their axons in the periphery. Interestingly, different types of defects are seen in the innervation of proximal versus distal leg muscles. In the innervation of the proximal femur muscles, PlexA/Sema1a knockdown results in phenotypes indicative of defective axon defasciculation and

decreased branching. In contrast, in the innervation of the distal tibia muscles, PlexA/Sema1a knock-down results in phenotypes indicative of increased axonal branching. Thus, both decrease and increase of motor axon branching can result if PlexA/Sema1a action in leg motoneurons is impaired. At the cellular level, both these phenotypes may be due to defects in early born femur motoneurons which manifest a loss of appropriate proximal femoral innervation and gain tibial innervation. In addition, increased arborization of targeted late-born tibial motoneurons in the distal tibia is also likely to contribute to the observed phenotypes. These findings suggest that the same Plexin/Semaphorin signaling system might have multiple roles in motor axon guidance. Since these different effects appear to be motoneuron-specific, they may be linked to motoneuron birth-order.

An interesting feature of all the PlexA/Sema1a phenotypes observed in our experiments is that they are the result of knockdowns limited to motoneurons, either in their ensemble or in specific small sets. In all cases, the motoneurons that manifest these peripheral axonal defasciculation and targeting defects are the only cells in the leg that have impaired levels of PlexA/Sema1a. In this respect, the requirement of both PlexA and Sema1 for correct axonal targeting is autonomous to leg motoneurons. This suggests that PlexA/Sema1a signaling might be at the level of axon-axon interactions. A role of Plexin/Semaphorin signaling in axon-axon interactions has been documented in the developing visual and olfactory systems of *Drosophila* (*Sweeney et al., 2007*; *Hsieh et al., 2014*). Alternatively, cis-interactions between PlexA and Sema1a within the affected motoneuron axons might also be involved. In murine models, cis- interactions between Sema6A and PlexA4 have been shown to modulate the repulsive response to Sema6A in sympathetic and sensory neurons (*Haklai-Topper et al., 2010*). While our experiments document an autonomous requirement of PlexA/Sema1a within the ensemble of leg motoneurons, they do not rule out the existence of PlexA/Sema1a signaling between outgrowing motor axons and their target muscles in the leg. Thus, it will be important to study both the expression and function of Plexin/Semaphorin signaling in the developing leg muscles.

## Plexins and Semaphorins in motoneuron dendrite targeting

During development, motoneurons must not only project their axons to the appropriate postsynaptic target muscles, they must also ensure that their dendrites, as major input sites, are positioned correctly in the neuropile in order to receive the appropriate presynaptic drive from pre-motor interneurons and sensory neurons (*Parrish et al., 2007*; *Vrieseling and Arber, 2006*; *Harris et al., 2015*; *Arber, 2012*). Although the mechanisms that control this type of targeted dendritogenesis are poorly understood, recent work on embryonic and adult-specific leg motoneurons in *Drosophila* indicate that their dendrites are organized topographically as myotopic maps that reflect their innervation pattern in the periphery (*Landgraf et al., 2003*; *Brierley et al., 2012*; *2009*; *Baek and Mann, 2009*).

For leg motoneurons, the majority of which are generated post-embryonically in a single neuroblast lineage, a birth-order dependent pattern of dendritic targeting and of peripheral innervation has been documented (*Brierley et al., 2009*; *2012*). Thus, early-born motoneurons, which innervate proximal muscle groups, position their dendrites in medial and lateral neuropile, while late-born motoneurons, which innervate distal muscle groups, elaborate their dendrites in the lateral neuropile only, and intermediate-born motoneurons target their dendrites to intermediate neuropile positions. Irrespective of their birth-order, all leg motoneurons initiate dendritogenesis synchronously and form their dendritic arbors in a targeted manner by growing into their appropriate area of innervation. Leg motoneuron dendrites attain their final architecture through targeted dendritogenesis. Apart from the intrinsic signaling cues, extrinsic cues could be implicated in this process. Although glia are known to secrete cues and neurotropic factors for axonal morphogenesis and survival (*Jacobs, 2000*; *Martin et al., 2012*; *Zlatic et al., 2009*; *Klämbt et al., 1991*), their role in regulation of dendritic morphology is relatively unexplored. While recent findings in vertebrates and invertebrates suggest that glia could play a role in dendritogenesis, the mechanism underlying these effects is not known (*Procko and Shaham, 2010*). Midline glia extend gliopodia during axonogenesis, and reducing their numbers also leads to the shift in neuropil during embryonic nervous system development in *Drosophila* (*Vasenkova et al., 2006*) suggesting that these could also play a role in dendritogenesis. Investigation of the molecular mechanisms involved in dendritic development of leg motoneurons has focused on the role of Slit/Roundabout and the Netrin/Frazzled (*Brierley et al., 2009*) signaling systems in controlling the precise targeting of motoneuron dendrites along the medio-lateral axis of

the ganglionic neuropile. While Slit/Netrin could be derived from midline glia, the role of other midline cells (*Kearney et al., 2004*) has not been ruled out. In this context, the substantially broader expression pattern of Sema2a, compared to that of Slit (which is restricted to the midline) is pertinent (*Figure 5B,E*; *Figure 5—figure supplement 1*). Dendritic targeting in larval motoneurons, which are of embryonic in origin is independent of glial differentiation (*Landgraf et al., 2003*; *Landgraf and Thor, 2006*). Our investigation of dendritic targeting of leg motoneurons in adult *Drosophila* provides further insight into the role of glia in the control of dendritogenesis. Moreover, they identify PlexB/Sema2a as additional signaling system that mediates this control. Our data are in accordance with a model in which Sema2a is secreted from midline and other glial cells and acts through the PlexB receptor to restrict dendrites of late-born motoneurons to appropriate lateral neuropile domains. Loss of PlexB in these motoneurons as well as loss of Sema2a from glial cells, as seen by using both Repo-Gal4 and Slit-Gal4 drivers, results in a medial shift in dendritic arbors together with the mis-projection of dendrites toward the midline. As a result, the affected dendrites of late-born neurons acquire the topographic morphology of normal early-born neurons in the ganglionic neuropile. Since cells at/and near the ganglionic midline express high levels of Sema2a, and since targeted knockdown of Sema 2a specifically in glial cells using Repo-Gal4 results in the mispositioning of late-born motoneuron dendrites, we postulate that glial cells provide the secreted Sema2a, which acts on dendrite targeting in late-born motoneurons. Thus, while the requirement of PlexB for dendrite targeting in late-born motoneurons is cell autonomous, the requirement of Sema2a is non-cell autonomous and presumably is provided through secretion from glial cells. We also observe a dorso-ventral gradient of Sema1a and Sema2a (*Figure 5—figure supplement 1*). This suggests a role in dorso-ventral patterning as well, an aspect which we have not examined.

## Multiple roles of Plexin/Semaphorin signaling systems in central and peripheral motoneuron circuitry development

Taken together, our findings indicate that two different Plexin/Semaphorin signaling systems act to play a range of roles in establishing the specific leg motoneuron architecture needed for appropriate motor circuitry. PlexA/Sema1a is required for correct peripheral axon guidance, and for both PlexA and Sema1a this requirement is motoneuron-specific. PlexB/Sema2a is required for correct central dendrite targeting, and while this requirement is motoneuron-specific for PlexB, the requirement for Sema2a in motoneurons is non-cell autonomous and likely involves glial cells. Interestingly, while PlexA/Sema1a appears to be required for both early-born and late-born motoneuron axon guidance, PlexB/Sema2a appears to be required for late-born, but not for early-born motoneuron dendritic patterning.

The relative independence of action of these two different Plexin/Semaphorin signaling systems in axon guidance and dendrite targeting of leg motoneurons suggests that both developmental processes could, in principle operate at separate times. This seems unlikely as timeline analysis indicates that both peripheral axonal outgrowth and central dendritic targeting of leg motoneurons occur at the same/overlapping times during postembryonic development (*Brierley et al., 2009*; *2012*). The molecular mechanisms that orchestrate this simultaneous formation of axonal and dendritic arbors are not known. Possible mechanisms include cell-intrinsic temporal modulation of transcription factors controlling guidance molecule expression as well as retrograde feedback from the innervated muscle targets as seen in the vertebrate spinal cord (*Vrieseling and Arber, 2006*; *Enriquez et al., 2015*).

Our behavioral experiments indicate that specific experimental perturbations in the expression levels of the Plexin/Semaphorin signaling systems result in abnormal walking behavior. Moreover, they show that the same molecular perturbations also result in defective axonal and dendritic branching and arborization patterns in leg motoneurons. Although other effects of the Plexin/Semaphorin signaling systems on the neural circuitry for walking control may exist, the striking correlation between perturbation in leg motoneuron architecture and defective walking behavior provides further support for the idea that peripheral and central architecture is likely to be important for the function of the motoneurons involved in the neural circuitry for walking.

Although we did not observe any defective footprint pattern using the soot plate assay, following Sema2a knockdown in midline/glia using Repo-Gal4 or mid-line specific Slit-Gal4 (data not shown), we do not rule out the possibility of other intricate walking parameters e.g. speed, stance, swing-duration or stance linearity etc. being affected in this case. Given the conservation of molecular cues

involved in the development of neuronal morphology and connectivity, further insight into the Plexin/Semaphorin-dependent mechanisms that operate in motoneuron development in *Drosophila* is likely to be of general importance for understanding the basis of circuit formation in all parts of the nervous system (*Pasterkamp, 2012*).

## Material and methods

### Fly strains and genetics

The following fly strains were used;

OK371-Gal4,UAS-mCD8GFP, UAS-dicer; OK371-Gal4,UAS-mCD8GFP, Repo-Gal4, Slit-Gal4, VGN9281-Gal4,UAS-mCD8GFP (*Brierley et al., 2009*), VGN6341-Gal4,UAS-mCD8GFP, R60B11-Gal4 (39238; Bloomington Drosophila Stock Center). RNAi lines and mutants used in these experiments have been described previously. UAS Sema1a RNAi (VDRC, TRiP, Gift from Alex Kolodkin (*Sweeney et al., 2007*), UAS PlexA RNAi (VDRC; Gift from Liqun Luo (*Sweeney et al., 2007*; *Pecot et al., 2013*), UAS Sema2a RNAi (VDRC (*Sweeney et al., 2011*), UAS PlexB RNAi (VDRC, TRiP), CA07125 (Fly trap), Sema1a P[1] and Sema1a P[2] (*Yu et al., 1998*), plexADf(4)C3 (PlexA[09]) (*Winberg et al., 1998*), plexB[KG00878] (*Ayoob et al., 2006*; *Bellen et al., 2004*).

### Generation of VGN 6341-Gal4 transgenic flies

A plasmid containing 505 base pair DVGlut gene enhancer that corresponds to the *Drosophila melanogaster* genome coordinates, BDGP6:2L:2394617:2395172 was created as follows: The nucleotide sequence was amplified from wild-type Drosophila melanogaster DNA to include restriction enzyme cleavage sites, BamH1 using the primers 5' EcoR1(+) TTTTCGCCTTTTTGCAGTC and 3'BamH1(+) GCTTCAGCAGCAAACAATGA. The pPT Gal-attB vector (*Brierley et al., 2009*) was modified to contain the previously amplified enhancer region cloned directionally into EcoR1 and BamH1 sites. This plasmid containing enhancer sequence was then injected into attP2 fly embryos (*Sharma et al., 2002*) to make VGN 6341-Gal4 transgenic line.

### Analysis of walking behavior

Glass slides (dimensions: 7 cm x 5 cm) were coated with a thin layer of carbon soot using a candle. Care was taken so that the soot layer was uniform. As the fly walks across the soot plate, soot is dislodged from the points of contact. Thus we obtain footprints of the fly, which can be analyzed further (as described in *Maqbool et al., 2006*). The soot plates were imaged under a trinocular transmission microscope using a digital camera (Canon Powershot). The images were converted to gray-scale, and the brightness and contrast was adjusted, for easy comparison. Further binary dilation was performed to improve visibility of footprints.

### Immunocytochemistry

Thoracic ganglia were dissected at different developmental stages in pupa and adult in phosphate buffer saline (pH 7.8) (PBS) and fixed in 4% buffered formaldehyde for 45 min at 4°C. Immunocytochemistry was performed according to *Brierley et al., 2009*.

The following primary antibodies were used: Chicken pAb α-GFP (1:1000, Abcam), Rabbit (Rb) α-GFP (1:1000, Abcam), mouse (ms) α-Neuroglian (BP104, 1:40; DSHB), ms α-Sema2a (1:10; DSHB), Rb α -Sema-1a (*Yu et al., 1998*) (1:3000), Rb α -PlexA (*Sweeney et al., 2007*)(1:500). Secondary antibodies (1:400) from Invitrogen conjugated with Alexa fluor-488, 568 and 647 were used in all staining procedures.

### Dye backfilling

Retrograde labeling of motoneurons was done by dye backfilling the specific leg muscles that are innervated by either early or late born motoneurons. A small crystal of the dye was inserted into the neuromuscular junction and it was allowed to diffuse for four hours at 4°C. To label the early born motoneuron, ltm2 muscle was backfilled. To label the late born motoneuron, distal muscle of the tibia (tadm/ tarm) was backfilled. Thoracic ganglia were dissected and fixed in 4% buffered formaldehyde for 45 min at 4°C. Following a wash with PTX, tissues were mounted on slides in Vectashield

(Vector labs). Alternatively after fixing the samples, immunolabeling was performed following the same protocol as mentioned above.

The following dyes were used from Invitrogen:

1. Dextran tetramethyl rhodamine, anionic, lysine fixable (DMR), 3000KDa
2. Dextran, Fluorescein, anionic, lysine fixable, 3000 KDa.

## Confocal microscopy and image processing

Fluorescently labeled thoracic ganglia and legs were imaged at 60X using Olympus FV 1000 confocal point scanning microscope. Optical sections were collected at 1 micron interval and imported into NIH Image J (http://rsb.info.nih.gov.nih-image/). For processing dendritic images, sensory axons arbors were removed using Lasso tool if required. The maximum z-projections were then imported into Photoshop (Adobe, San Jose, CA) and minor adjustments were made to the brightness and contrast when required. To view axonal arborization of the motoneurons, we directly imaged GFP labeled motoneurons through the body wall and leg of adult flies. For leg images, auto-fluorescence from the cuticle that marked the outline was removed from each optical section using Lasso tool. The 3D reconstructions were generated using Amira (Visage Imaging, Berlin, Germany). To quantify the distribution of the Sema 1a and Sema2a along the medio-lateral axis of the neuropil, we used the plot profile tool in ImageJ. Analysis was performed as described in *Brierley et al., 2009*.

## Acknowledgements

We thank Alex L Kolodkin, Liqun Luo, S Lawrence Zipursky, Gerald M. Rubin, the VDRC, the Bloomington Stock Centre and the NCBS fly facility for generously providing fly stocks and other reagents. We thank Sudhir Palliyil, Akila Sridar and Pushkar Paranjpe for standardizing and pioneering the assays used to study walking behavior and Kirti Rathore for sharing her observations on motoneuron development with us. We thank Mani Ramaswami and Vatsala Thirumalai for helpful comments and discussions. We would also like to thank Centre for Nanotechnology, NCBS (Department of Science and Technology; Grant No. SR/S5/NM- 36/2005), for the Olympus FV1000 microscopes in the Central Imaging and Flow facilities at NCBS. This paper is dedicated to the memory of Veronica Rodrigues and KS Krishnan, whose collaboration and mentorship provided valuable insights into this study.

## Additional information

### Competing interests

KV: Senior editor, *eLife*. The other authors declare that no competing interests exist.

### Funding

| Funder | Author |
| --- | --- |
| NCBS-TIFR | Durafshan Sakeena Syed<br>Swetha B.M. Gowda<br>O Venkateswara Reddy<br>K VijayRaghavan |
| Swiss National Science Foundation | Heinrich Reichert |
| JC Bose Fellowship And CEFIRA | K VijayRaghavan |

The funders had no role in study design, data collection and interpretation, or the decision to submit the work for publication.

### Author contributions

DSS, Conception and design, Acquisition of data, Analysis and interpretation of data, Drafting or revising the article; SB.MG, Conception and design, Acquisition of data, Analysis and interpretation

of data; OVR, Generation of VGN 6341-Gal4 transgenic flies, Acquisition of data, Contributed unpublished essential data or reagents; HR, KV, Conception and design, Analysis and interpretation of data, Drafting or revising the article

**Author ORCIDs**

K VijayRaghavan, http://orcid.org/0000-0002-4705-5629

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
