## [Decision Letter]

Thank you for submitting your work entitled "Glial and neuronal Semaphorin signaling instruct the development of a functional myotopic map for *Drosophila* walking" for consideration by *eLife*. Your article has been favorably evaluated by Janet Rossant (Senior editor) and three reviewers, one of whom is a member of our Board of Reviewing Editors.

The reviewers have discussed the reviews with one another and the Reviewing editor has drafted this decision to help you prepare a revised submission.

Summary:

This is a beautifully written manuscript focusing on the molecular mechanisms that define the myotopic map in the *Drosophila* neuromuscular system. In brief, Syed et al. identified a role for Sema/Plexin signaling in adult *Drosophila* motorneuron (MN) development by RNAi screening and behavioral gait analysis. Sema1/PlexA control innervation of proximal and distal leg muscles in a cell autonomous manner, Sema2a is likely secreted by midline glia and specifically affects late born MNs via neuronal PlexB. The authors show that Sema2a is highly expressed in the midline, thus preventing extension of dendritic arbors to medial positions. The data are considered to be uniformly high quality and sufficiently innovative and of broad enough appeal to justify publication in *eLife*.

There are two areas that the reviewers consider necessary to unquestionably substantiate the author's major conclusions. First, there were concerns regarding the reliance on RNAi. All reviewers acknowledge the challenges that are faced by the authors, working with genes on the 4th chromosome and there was considerable dialogue on how best to substantiate these data within a reasonable timeline. Ideally, the authors would be able to use MARCM, allowing genetic analysis within small numbers of cells. However, it was acknowledged that this may not be a reasonable request due to the position of the genes on the 4th chromosome. Alternative controls are suggested below. Second, there was discussion about bringing the level of resolution down to individual or small numbers of axons, and extending the quantitative analysis of dendrite morphology.

1) All reviewers shared a concern about the reliance on RNAi-based phenotypes. It would be highly advisable to include some controls and additional allelic combinations to corroborate these interesting findings.

1A) The authors used Sema1a alleles and Sema1a/PlexA trans-heterozygous animals to corroborate the observed axonal phenotypes, which is helpful. Additional controls could be either rescue with an RNAi resistant transgene, antibody staining showing that RNAi indeed affects Sema1a/2a levels specifically, or the use of additional trans-heterozygous allelic combinations (if available). Since the authors have shown they can detect endogenous Sema1a/2a they could use quantitative immunohistochemistry to confirm the specificity of the RNAi knockdown.

1B) For the PlexB/Sema2a data, it is suggested that the authors look at phenotypes in trans-heterozygous mutant animals and Sema2a immunostainings in the glial Sema2a RNAi knockdown. It might also be feasible to overexpress Sema1a in the midline and look at early born MNs if they are in this case repelled and look more like late born MNs.

2) The dendrite phenotype due to glial Sema2a or neuronal PlexB knockdown is striking and interesting. Again, an allelic combination of Sema2a/PlexB would be helpful to solidify the phenotype. A more quantitative assessment of the MN dendrite phenotype caused by Sema2a/PlexB knockdown would also be desirable (e.g. total dendrite length vs abnormal branch extending towards the midline).

3) It would be an important extension to bring the level of analysis down to single cell resolution or sparse labeling. This could be achieved using GAL4 lines with appropriately sparse labeling in order to refine the analysis regarding over/underinnervation of early vs. late born MNs.

---

## [Author Response]

There are two areas that the reviewers consider necessary to unquestionably substantiate the author's major conclusions. First, there were concerns regarding the reliance on RNAi. All reviewers acknowledge the challenges that are faced by the authors, working with genes on the 4th chromosome and there was considerable dialogue on how best to substantiate these data within a reasonable timeline. Ideally, the authors would be able to use MARCM, allowing genetic analysis within small numbers of cells. However, it was acknowledged that this may not be a reasonable request due to the position of the genes on the 4th chromosome. Alternative controls are suggested below. Second, there was discussion about bringing the level of resolution down to individual or small numbers of axons, and extending the quantitative analysis of dendrite morphology. 1) All reviewers shared a concern about the reliance on RNAi-based phenotypes. It would be highly advisable to include some controls and additional allelic combinations to corroborate these interesting findings. 1A) The authors used Sema1a alleles and Sema1a/PlexA trans-heterozygous animals to corroborate the observed axonal phenotypes, which is helpful. Additional controls could be either rescue with an RNAi resistant transgene, antibody staining showing that RNAi indeed affects Sema1a/2a levels specifically, or the use of additional trans-heterozygous allelic combinations (if available). Since the authors have shown they can detect endogenous Sema1a/2a they could use quantitative immunohistochemistry to confirm the specificity of the RNAi knockdown.

Experiments done: To address the concern 1A, we have tested another trans- heterozygous allelic combination of Sema1a (Sema1a^P1^/Sema1a^P2^) (Figure 4) to corroborate our findings. The phenotype observed is similar to Sema1a homozygous animals (Sema1a^P2^/Sema1a^P2^; Figure 4), and the trans-heterozygous allelic combination (Sema1a^P2^/PlexA; Figure 4) and RNAi knockdowns (Figure 2, Figure 3, Figure 4), where defective motor neuron axonal branching is observed. Decreased defasciculation and branching is seen in the proximal femur and increased axonal branching is observed in motor neurons that innervate tibia.

Furthermore, over expression of Sema1a and PlexA in both Femur (Figure 2) and tibia (Figure 3) show phenotypes ‘opposite’ to that seen in knockdowns and mutants. Finally, we have used multiple RNAi’s that target different regions of Sema1a and Plex A. All of them show similar phenotype upon RNAi induced knockdown in motor neurons.

1B) For the PlexB/Sema2a data, it is suggested that the authors look at phenotypes in trans-heterozygous mutant animals and Sema2a immunostainings in the glial Sema2a RNAi knockdown. It might also be feasible to overexpress Sema1a in the midline and look at early born MNs if they are in this case repelled and look more like late born MNs.

Experiments done: To address the concern 1B, we performed the following experiments;

i) We tested the dendritic targeting in Sema2a/PlexB trans-heterozygous mutant animals (Figure 6—figure supplement 2). Late-born motor neurons that normally restrict their dendritic arbors towards the lateral regions of the ganglionic neuropile, mis-project their dendrites towards the midline. The phenotype is similar to Sema2a knockdown in glia or Plex B knockdown in motor neurons (Figure 6, Figure 6—figure supplement 1). The Sema2a RNAi has been widely used by others and has been shown to reduce levels of Sema2a (Sweeney et al., Neuron, 2011).

ii) We over-expressed Sema1a in midline and looked at the dendritic arborisation of early born-motor neurons. These neurons normally project a major dendritic branch towards the midline. Following over-expression of Sema1a or Sema2a at the midline using midline specific Slit-Gal4 driver, we did not observe any effect on the dendritic arborisation of early born motor- neurons (data not shown).

2) The dendrite phenotype due to glial Sema2a or neuronal PlexB knockdown is striking and interesting. Again, an allelic combination of Sema2a/PlexB would be helpful to solidify the phenotype. A more quantitative assessment of the MN dendrite phenotype caused by Sema2a/PlexB knockdown would also be desirable (e.g. total dendrite length vs abnormal branch extending towards the midline).

Experiments done:Sema2a/PlexB trans-heterozygous allelic combination was tested as explained above (Figure 6—figure supplement 2). The phenotype is the extension of major primary dendritic branch towards the midline which is never observed in controls. This is striking that and is a Yes/No phenotype. It should be noted that in dye backfilling all secondary branches are not marked, so quantification in terms of pixel intensity will give variation and lead to artifacts.

3) It would be an important extension to bring the level of analysis down to single cell resolution or sparse labeling. This could be achieved using GAL4 lines with appropriately sparse labeling in order to refine the analysis regarding over/underinnervation of early vs. late born MNs.

Experiments done: To address concern 3, we had already shown, using the VGN6341-Gal4 driver that marks just 3-4 motor neurons that innervate femur (early born motor neurons) and 1-2 motor neurons that innervate tibia (late-born motor neurons) (Figure 4). However, in addition we have now used R60B11-Gal4 that marks 3-4 motor neurons that innervate femur and 2-3 motor neurons that innervate tibia (Figure 4—figure supplement 1). We also used VGN 9281-Gal4 driver that marks a single motor neuron that innervates distal most muscle of the tibia (Figure 4—figure supplement 2).